# The Effectiveness of Artificial Intelligence in Assisting Mothers with Assessing Infant Stool Consistency in a Breastfeeding Cohort Study in China

**DOI:** 10.3390/nu16060855

**Published:** 2024-03-15

**Authors:** Jieshu Wu, Linjing Dong, Yating Sun, Xianfeng Zhao, Junai Gan, Zhixu Wang

**Affiliations:** 1Department of Maternal, Child and Adolescent Health, School of Public Health, Nanjing Medical University, Nanjing 211166, China; jwu@njmu.edu.cn (J.W.); linjingdong2022@163.com (L.D.); yatingsun2024@163.com (Y.S.); 2Danone Open Science Research Center, Shanghai 201204, China; xianfeng.zhao@danone.com (X.Z.); junai.gan@danone.com (J.G.)

**Keywords:** stool consistency, CNN algorithm, postpartum depression status, breastfed infants

## Abstract

Breastfeeding is widely recognized as the gold standard for infant nutrition, benefitting infants’ gastrointestinal tracts. Stool analysis helps in understanding pediatric gastrointestinal health, but the effectiveness of automated fecal consistency evaluation by parents of breastfeeding infants has not been investigated. Photographs of one-month-old infants’ feces on diapers were taken via a smartphone app and independently categorized by Artificial Intelligence (AI), parents, and researchers. The accuracy of the evaluations of the AI and the parents was assessed and compared. The factors contributing to assessment bias and app user characteristics were also explored. A total of 98 mother–infant pairs contributed 905 fecal images, 94.0% of which were identified as loose feces. AI and standard scores agreed in 95.8% of cases, demonstrating good agreement (intraclass correlation coefficient (ICC) = 0.782, Kendall’s coefficient of concordance W (Kendall’s W) = 0.840, Kendall’s tau = 0.690), whereas only 66.9% of parental scores agreed with standard scores, demonstrating low agreement (ICC = 0.070, Kendall’s W = 0.523, Kendall’s tau = 0.058). The more often a mother had one or more of the following characteristics, unemployment, education level of junior college or below, cesarean section, and risk for postpartum depression (PPD), the more her appraisal tended to be inaccurate (*p* < 0.05). Each point increase in the Edinburgh Postnatal Depression Scale (EPDS) score increased the deviation by 0.023 points (*p* < 0.05), which was significant only in employed or cesarean section mothers (*p* < 0.05). An AI-based stool evaluation service has the potential to assist mothers in assessing infant stool consistency by providing an accurate, automated, and objective assessment, thereby helping to monitor and ensure the well-being of infants.

## 1. Introduction

Breast milk provides newborns with essential bioactive factors for immune maturation and healthy microbial colonization [1], which are beneficial for infant gastrointestinal health. Its composition, rich in lactose and unique fats, is more easily digestible and absorbable than formula milk [2,3]. When lactose is not fully absorbed in the small intestine, it ferments in the colon, increasing stool water content and making it softer [4]. Additionally, breast milk’s oligosaccharides enhance beneficial gut bacteria, such as Bifidobacteria and Lactobacillus, further influencing stool consistency through fermentation and water-content regulation [5]. Breastfed infants also frequently experience “minor” gastroenterological indications and symptoms that need to be recognized early [6], which are usually signaled initially by stool characteristics. However, infants’ feces differ considerably from those of adults, being small in quantity and frequently unshaped [7]. Recognizing the fecal characteristics of breastfeeding infants is crucial for inexperienced parents to distinguish normal infant stool patterns and prevent unnecessary concerns, delayed medical treatment, and additional healthcare expenses [8].

Stool consistency reflects fecal water content and total bowel transit time [9] and is related to species richness and gut microbiota community composition [10]. In pediatric gastroenterology, stool consistency represents the defecation pattern of children and aids in the diagnosis of functional gastrointestinal disorders [11]. For clinical and parental use, the Stool Consistency Scoring Tool was developed to help describe and differentiate between physiological and pathological stool appearance, such as the Bristol Stool Scale (BSS) [9,12], Amsterdam Infant Stool Scale (AMS) [13,14], and Brussels Infant and Toddler Stool Scale (BITSS) [15,16]. The BSS was suggested by Rome IV and is widely used in adult and pediatric clinical diagnostics and trials; however, it is not considered appropriate for young children [17]. Subsequently, the AMS [13,14] and BITSS [18,19] were developed as scales for assessing stool consistency in non-toilet-trained children, and their validity for feces in diapers was verified [16], including a related Chinese version [19]. These scoring techniques may appear simple and uncomplicated, but they are influenced by subjective evaluations, particularly by inexperienced parents or in situations of cognitive bias caused by psychological states [20]. Early infants, especially breastfed infants, have softer and smaller amounts of stool [13], which are thick or thin, and usually have curds [7]. Stools are spread out in the diaper and pressed together between the buttocks, adding to the difficulty and distress of manual scoring for mothers.

The application of artificial intelligence (AI) in medical image processing, including convolutional neural network (CNN) algorithms, assists in disease screening and diagnosis in various clinical settings [21,22,23,24]. Previously, machine learning and CNN algorithms were integrated with a smartphone application for automated digital image assessment of stool consistency in diapers [25,26]. However, previous research did not focus on breastfeeding infants to address the specific challenges in the evaluation of their stool consistency. In this study, we applied an AI-based stool evaluation service in an observational cohort study, assessed the AI-graded scores as well as the mother-reported scores, and identified those who could benefit most from this AI service.

## 2. Materials and Methods

### 2.1. Schematic Overview of the Cohort Study

An observational cohort study was conducted to explore the relationships between breastfeeding, breast milk composition, and infant development and health. Approximately 750 pairs of mothers and infants were recruited from six different cities in China, following up at 1, 4, 6, and 12 months of infant age. This paper uses data from the Nanjing site at the first visit to understand the applicability of a stool photographic AI tool in one-month-old infants and the benefits to the mothers. The research process is shown in Figure 1.

### 2.2. Study Participants

A total of 131 mother–infant pairs were recruited by maternal and child health professionals in Jiangning District, Nanjing, between November 2021 and September 2022. The inclusion criteria were as follows: (1) Mothers aged 18 years or older, with a pre-pregnancy or first routine pregnancy check-up BMI of 18.5–28; (2) full-term infants (gestational age 37–42 weeks); and (3) mothers willing to breastfeed, and infants being breastfed at enrollment (including exclusive breastfeeding or supplemented with formula). The exclusion criteria were as follows: (1) Mothers who conceived twins, multiples, or through assisted reproductive technologies; (2) infants who consumed anything other than breast milk at the time of enrollment, such as infant formula and water; (3) mothers with specific diseases, such as severe illnesses, psychiatric disorders, moderate postpartum depression, or mastitis; (4) infants with congenital anomalies, chromosomal disorders, or serious illnesses; and (5) mothers or infants participating in other interventional studies.

All individuals agreed to participate in the study and signed the informed consent forms. This study was approved by the Nanjing Medical University Ethics Committee (2021-616).

### 2.3. Site Investigation and Quality Control

Face-to-face interviews were conducted to collect information on maternal sociodemographic characteristics, infant gastrointestinal symptoms, infant quality of life, and maternal postpartum depression status. Prenatal and delivery records from healthcare handbooks and hospital discharge records were used to determine gestational history, mode of delivery, and pregnancy-related problems. This was carried out to guarantee data validity and accuracy. The surveys were conducted in-home by professionally trained researchers, and on-site inspections were performed to avoid missing or logically erroneous data.

### 2.4. Assessment of Infant Gastrointestinal Symptoms, Quality of Life, and Maternal Postpartum Depression

The Infant Gastrointestinal Symptom Questionnaire (IGSQ) was used to evaluate gastrointestinal tolerance in infants [27]. The 13-item IGSQ divides questions into five categories: stool characteristics, vomiting, crying, fussiness, and bloating. Each item is scored on a scale of 1 to 5, and the total score ranges from 13 to 65. Higher scores indicate more severe symptoms.

The Pediatric Quality of Life Inventory (PedsQL) was used to assess infant quality of life [28,29]. The 36 questions covered 5 dimensions: physical functioning, physical symptoms, emotional functioning, social functioning, and cognitive functioning. Each item is scored on a 5-point Likert scale ranging from 0 (never) to 4 (almost always). Higher scores correspond to a higher quality of life in relation to health. The average of all the question scores determined the final score. The psychosocial health score is the average score for questions about emotional, social, and cognitive functioning, and the physical health score is the average score for questions on physical functioning and physical symptom scales.

The Edinburgh Postnatal Depression Scale (EPDS) was used to assess the risk of postpartum depression [30]. The EPDS comprises 10 items in total: mindfulness, happiness, self-blame, depression, anxiety, insomnia, coping skills, sadness, crying, and self-harm. Four response alternatives are provided for each issue, representing different levels of symptom severity, from “never” to “always”, with scores ranging from 0 to 3. An individual’s total score, which ranges from 0 to 30, is calculated by adding the values for each of the 10 items. Higher scores indicated more severe depression in this study, where EPDS ≥ 10 was considered postpartum depression (PPD).

### 2.5. Photographic Documentation and Evaluation of Diaper Feces

Participants were required to take photographs of their infant’s feces on diapers using their smartphones at home and upload them to an applet (Stool Tracker v2.0). The applet initially identified the diaper backdrop in the photographs upon upload. Users were prompted to retake or upload a new photograph if the app failed to identify it. The researchers provided the participants with detailed instructions on how to take and upload the photographs.

The Brussels Infant and Toddler Stool Scale (BITSS) [11] is a visual scale used to assess stool consistency in non-toilet-trained children. It consists of seven photographs of various stool types grouped into four categories: hard (Types 1–3), formed (Type 4), loose (Types 5 and 6), and watery (Type 7). Types 1–7 of the four categories correspond to scores 1–7, respectively. A stool image recognition algorithm based on deep convolutional neural networks was developed according to the BITSS [25]. Furthermore, the algorithm was upgraded in China using larger datasets and achieved 92.9% accuracy for seven BITSS types and 95.4% accuracy for four BITSS categories [26]. The upgraded version of the algorithm was used in this study and can automatically score each stool image and assign the BITSS type and category (AI scores).

Upon uploading infant stool photographs, mothers were also required to rate stool consistency using the same scoring criteria for the four types (mothers’ scores). Two trained independent researchers assigned the stool images to seven types and four categories with the same BITSS scales. When there was a difference in scoring between the two researchers, the images were reassessed until a consistent result was obtained (standard scores). Neither of the researchers had access to information on the participants.

### 2.6. Consistency Evaluation of Photographs of Baby Diaper Feces

The AI and mothers’ scores were compared with the standard score using agreement, intraclass correlation coefficients (ICCs), and Kendall’s coefficient of concordance W (Kendall’s W) for consistency evaluation of fecal photographs. Agreement was calculated as the percentage of photographs with matched scores. A mean-rating, absolute-agreement, 2-way random-effects model was used to calculate ICC values. ICC values < 0.5 indicate poor agreement, 0.5–0.74 suggest moderate agreement, 0.75–0.9 show good agreement, and >0.9 indicate excellent agreement. Kendall’s W was used to measure the degree of agreement in stool consistency between AI, mothers, and standard scores [31]. Based on the value of Kendall’s W, agreement was classified as poor (Kendall’s W ≤ 0.00), slight (0.01 ≤ Kendall’s W ≤ 0.20), fair (0.21 ≤ Kendall’s W ≤ 0.40), moderate (0.41 ≤ Kendall’s W ≤ 0.60), good (0.61 ≤ Kendall’s W ≤ 0.80), or excellent (0.81 ≤ Kendall’s W ≤ 1.00). Kendall’s tau was used to evaluate the correlation of per-photograph scores between the AI, mothers, and researchers.

The degree of inconsistency was also evaluated. In cases of conflict, the mean level of disagreement for each evaluable photograph was calculated and defined as the mean difference in scores. The deviation degree of the AI score and mothers’ score from the standard score was established by calculating the absolute value of the difference and dividing this difference by the number of assessments. Subsequently, the degrees of deviation between the two were compared.

### 2.7. Statistical Analysis

An independent sample *t*-test was used to compare normally distributed continuous variables reported as the mean ± standard deviation (*x* ± SD) between photograph-upload and non-upload groups. The rank sum test was used to compare non-normally distributed continuous variables expressed as medians (P25, P75) between groups, using the Mann–Whitney U test for two-group comparisons and the Kruskal–Wallis H test with Bonferroni’s correction for three-group comparisons. The Chi-squared (*χ*^2^) test was used to assess categorical data, which were reported as frequency (n) and percentage (%). To identify the factors impacting the degree of divergence in mothers’ scores, a Tobit regression model was used. SPSS (version 26.0) and R 4.3.2 were used to analyze and process the data. A *p*-value of 0.05 on both sides was considered statistically significant.

## 3. Results

### 3.1. Characteristics of the Study Population

A total of 131 mother–infant pairs were included in this study. The mothers in the study had an average age of 31.6 ± 3.5 years, with 10.7% being housewives and 32.8% having education below university level. Of the infants, 55.0% were fed breast milk only. The EPDS had a median score of 7 points. Detailed demographic information for the study participants is shown in Table 1.

Of the 131 mother–infant pairs, 98 uploaded fecal images, and 33 did not. Between-group comparisons of the basic characteristics of the two groups were performed (Appendix A). The proportion of cesarean sections in the uploaded photographs group was significantly lower (*p* < 0.05), and there was no statistically significant difference between the two groups in terms of delivery time or history of pregnancy complications (*p* > 0.05).

In addition, comparison of IGSQ and PedsQL scores between the two groups showed that the IGSQ total, crying, and fussiness scores of the uploaded photographs group were significantly higher than those of the no uploaded photographs group; the total, physiological function and physical health scores of PedsQL were significantly lower than those of the no uploaded photographs group, and the difference was statistically significant (*p* < 0.05) (Appendix A).

### 3.2. Comparison of Mothers’ Self-Reported Scores and AI-Graded Scores

Using the WeChat applet, 974 stool photographs of 98 one-month-old infants were collected. Of these, 69 photographs were excluded owing to duplication. Ultimately, 905 stool photos were included in the analysis. As determined by the researchers, the distribution of submitted photos across the seven BITSS types was 5 (0.6%), 851 (94.0%), and 49 (5.4%) for Types 5, 6, and 7, respectively. A confusion matrix predicted by the AI for fecal photographs is shown in Appendix A. There were 856 (94.6%) loose and 49 (5.4%) watery stool samples when the seven BITSS types were categorized into the four classes determined in the validation study. There were no differences in the consistency of infant feces between the breastfeeding and mixed-feeding groups (*p* = 0.552).

AI and standard scoring exhibited a good correlation not only in seven types (ICC = 0.754, Kendall’s W = 0.836, with a 95.5% agreement) but also in four categories (ICC = 0.782, Kendall’s W = 0.840, with a 95.8% agreement) (Table 2). The AI scores were 0.019 points higher than the standard scores in seven grades and 0.017 points higher in four grades on average.

Mothers scored in only four categories. Consistency was poor between mothers’ assessments and standard scorings (ICC = 0.070, Kendall’s W = 0.523, with a 66.9% agreement) (Table 2). Mothers’ scores of images were 0.260 points higher than those of the researchers. Figure 2 shows greater disagreement with mothers’ evaluations than those of the AI. Among 856 samples rated as loose stools, the agreement between mothers’ scores and standard scores was 68.3% (95% CI 63.7%, 72.9%); among the 49 items evaluated as watery stool types, the exact agreement was 42.9% (95% CI 29.0%, 56.8%). Kendall’s tau coefficients of 0.690 between AI and standards and 0.058 between parents and standards were observed (Figure 3).

Additionally, for AI deviation, the median and interquartile range were 0.00 (0.00, 0.10), while for mothers, they were 0.18 (0.00, 0.69). Notably, the deviation levels of mothers were substantially higher than those generated by the AI (*p* < 0.001).

### 3.3. Analysis of Factors Influencing the Degree of Mothers’ Deviation

Table 3 shows a higher degree of deviation among mothers with an education level of junior college and below (*p* < 0.05) and a significant correlation between the degree of mothers’ deviation and EPDS scores (*p* < 0.05). It seemed that mothers who were unemployed, had a cesarean delivery, or with a tendency toward postpartum depression also had a higher degree of deviation (*p* < 0.1). No significant differences in deviations were found among mothers with different income levels, parity, delivery methods, or feeding methods.

Based on the four characteristics that may lead to maternal assessment deviation (*p* < 0.1), the participants were categorized into three groups: those without any identified features, those with one feature, and those with two or more features. The degrees of deviation among the above three groups were 0.00 (0.00, 0.28), 0.18 (0.00, 0.73), and 0.27 (0.00, 0.94), respectively, which exhibited significant differences overall (*p* = 0.044) and significantly lower deviation in the group with no features compared with the group with two or more features (*p* = 0.013) (Figure 4).

### 3.4. Multivariable Tobit Regression Analysis

A Tobit regression model was used for subsequent multivariable analysis, with occupation, education level, mode of delivery, and EPDS score as independent variables and degree of deviation as the dependent variable (Table 4). The results revealed that for every increase in EPDS score, the deviation increased by 0.023 points. Education level (*β* = −0.149, *p* = 0.248), mode of delivery (*β* = 0.151, *p* = 0.234), and occupation (*β* = 0.205, *p* = 0.262) had no significant impact on the degree of mothers’ deviation.

We further stratified the Tobit regressions using variables that were insignificant in the multivariable model to explore the EPDS (Table 5) considering the higher degree of deviation in mothers with multiple characteristics. In the cesarean section group, for every increase in EPDS score, the deviation increased by 0.040 points. In the employed group, for every increase in the EPDS score, the deviation increased by 0.024 points. In other conditions, no association was found between EPDS scores and maternal deviation. The detailed results of the stratified Tobit models are presented in Appendix A.

## 4. Discussion

In this study, we first used the AI-based stool evaluation service in an observational cohort study specifically to evaluate the stool consistency of Chinese breastfed infants and its effectiveness in assisting mothers’ assessment. The present study found no significant difference in stool consistency between exclusively breastfed and mixed-feeding infants at 1 month of age. The CNN-based automated assessment of infant stool consistency was significantly more accurate than mothers, with an accuracy of 95.8% vs. 66.9% (Refer to Table 2). Mothers who had a cesarean section, were unemployed, had a lower education level, or had a tendency toward postpartum depression appeared more prone to inaccurate evaluation. The deviation increased by 0.023 points (*p* < 0.05) for each point increase in the EPDS score, which was present in mothers who were employed or had undergone a cesarean section (*p* < 0.05) (Refer to Table 4). Additionally, this study found that mothers who delivered vaginally or had infants with a lower quality of life or gastrointestinal problems were more likely to use this applet.

In the realm of medical image identification, the widespread use of AI algorithms based on the capacity of CNNs to differentiate minute details provides a powerful tool for automatic and impartial evaluation of the consistency of infant diaper stools. This study found that the predominant consistency in one-month-old infants was loose stools, which is consistent with a previous report on infants of the same age [26]. The moisture in the stool is readily absorbed by the diaper, and the assessment of stool consistency must be judged in conjunction with the moisture marks on the diaper. In previous studies, CCN-based AI has proven its ability to evaluate fecal consistency, with an agreement of 60–95% [25,26]. The agreement rates in our study were consistent with data published in the literature. In contrast to the AI, the mothers’ scores were consistently lower (agreement rate 66.9%, Kendall’s W = 0.523, ICC = 0.070, and Kendall’s tau = 0.058) (Refer to Table 2, Figure 3) in our cohort, which was slightly higher than 48.5~58% in a comparable BITSS study [25] and a BSFS study [32]. These results demonstrate the ability and the advantage of the CNN-based AI method to assess stool consistency.

In this study, seven types and four categories of the BITSS classification were used, and the fecal types were found to be primarily concentrated in Types 6 and 7 owing to the young age of the babies, which corresponded to classes 3 and 4 of the four categories. We found that mothers tended to score the images as watery stool (0.260 points higher than the researchers), and in the case of a mother’s error in assessment, a higher proportion of Type 3 feces were misclassified as Type 4, whereas only a minor percentage of Type 4 stools were classified as Type 3 with the AI method (Refer to Figure 2). This illustrates the benefits of AI-assisted home assessment of a baby’s stool consistency, which minimizes the misclassification of normal feces as watery feces.

Investigation of the factors determining the degree of manual assessment bias further suggests the clinical relevance of this AI-assisted method for practical implementation. We first found clues into the factors leading to mothers’ deviations through a comparison study. Employment status, mode of delivery, level of education, and risk of postpartum depression may have influenced the degrees of bias in mothers’ scores (Refer to Table 3). However, in the Tobit regression model, which was used for subsequent multivariable analysis owing to the presence of truncated and duplicate values at 0 in the data, only the EPDS score had an independent and significant effect on the degree of bias, and individuals with a higher risk of depression had a greater degree of assessment bias (Refer to Table 4). PPD is one of the most common postpartum disorders, with a prevalence of 17.2% worldwide [33] and 14.8% in China [34]. Cognitive biases are common among individuals with depression. Mothers suffering from depression and anxiety are more likely to perceive negative emotions (i.e., sadness) in their infants’ faces and engage in biased processing [20]. This is supported by mood perception studies showing that depressed people are more likely to turn their attention to negative faces [35]. This may explain why a larger proportion of mothers in our study categorized the normal stool into thinner (i.e., abnormal) groups. Many studies showed that the mode of delivery, mainly cesarean delivery, has a significant effect on the occurrence of postpartum depression [36,37]. According to a network meta-analysis, the risk of postpartum depression is approximately 1.5 times higher after cesarean delivery than after vaginal delivery [36]. Depression risk is also linked to characteristics such as unemployment and lower education levels [38,39]. We obtained similar results: cesarean section mothers had higher rates of depression than vaginal-delivery mothers (13/33 = 0.39 vs. 17/65 = 0.26, *p* = 0.183), and unemployed mothers had higher rates of depression than working mothers (7/14 = 0.5 vs. 36/117 = 0.31, *p* = 0.226), but these differences were not significant. Nevertheless, interestingly, we discovered that depression-induced cognitive bias may be more pronounced among cesarean section or employed mothers, suggesting that there are complicated interactions between the above-mentioned elements that need further exploration in larger groups (Refer to Table 5). Our finding that the degree of bias was substantially higher for those with multiple characteristics than for those with a single characteristic also supports these points (Refer to Figure 4).

Half of newborns, including breastfed infants, experience gastrointestinal symptoms, with only a small percentage requiring hospitalization [6]. This can be aided by the detection of the stool form, which has been shown to correlate closely with whole-gut transit time and was used in clinical practice and research [40]. The results of this study showed that the AI assessment method yielded more accurate results than the mothers’ assessments (Refer to Table 2). The more typical assessment bias of impressing the normal type (Category 3) over the polyhydric type (Category 4) can largely be avoided to serve those in need, especially mothers with depressive tendencies. Additionally, mothers who had a vaginal delivery or who had an infant with a lower quality of life or gastrointestinal problems were more likely to use this app to upload fecal photographs (Refer to Appendix A). This may be because mothers who had cesarean deliveries were in recovery in the first month after delivery and did not have the time to care about their infant’s fecal characteristics or take pictures, whereas mothers who perceive that their child’s quality of life is low or that their gastrointestinal symptoms are severe will want medical help and, therefore, are more likely to prioritize taking pictures to illustrate their infant’s stool characteristics [41]. This is one of the populations targeted by this method. The combination of deep network learning techniques with mobile devices such as smartphones has the potential to broaden the scope of work from clinical diagnosis to home care, allowing for low-cost universal diagnostic care.

Our study had certain limitations. First, the study was limited to parents who uploaded photographs, which may have resulted in incomplete samples and selection bias. Second, the study only included one-month-old infants whose feces were overwhelmingly loose and watery. The potential for an effective stool consistency assessment for a broader range of fecal types and more scenarios should be explored.

## 5. Conclusions

Trained CNN-based AI evaluations yielded automated and accurate assessments of stool consistency in breastfed infants, which were more reliable than mothers’ evaluations of infant stool. The AI-based stool evaluation service may be useful in clinical studies and home assessments to provide accurate and objective results on infant stool consistency. Further work is needed to evaluate the applicability and effectiveness of this AI service in a broader population and in more complex feeding situations.

## Figures and Tables

**Figure 1 nutrients-16-00855-f001:**
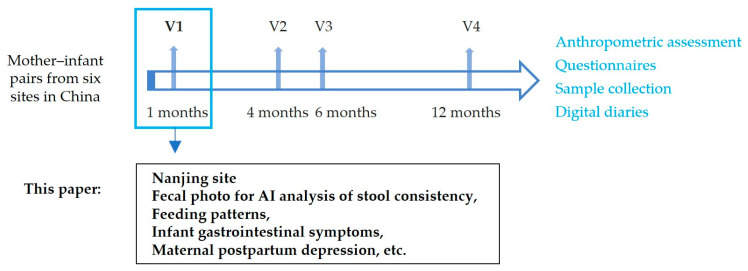
Research design process diagram.

**Figure 2 nutrients-16-00855-f002:**
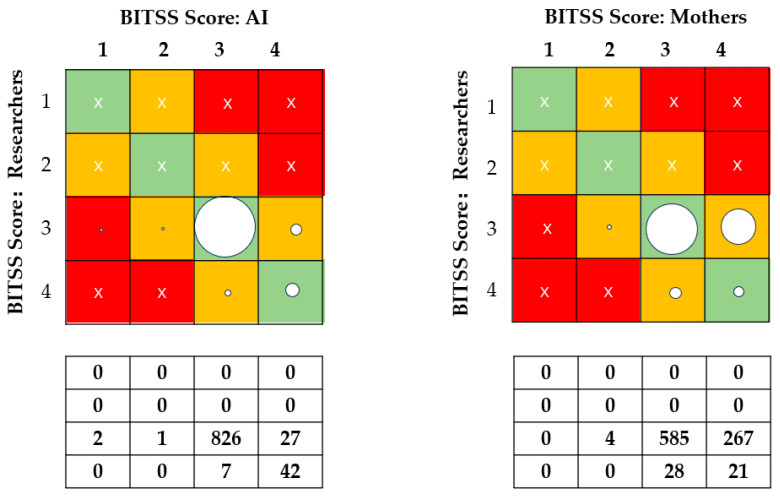
Proportions of exact agreement between AI, mothers, and researchers. BITSS = Brussels Infant and Toddler Stool Scale. Green cells: allocation matches the reference BITSS type for the corresponding photograph. Orange cells: allocation deviates by 1 level from the reference BITSS type for the corresponding photograph. Red cells: allocation deviates by more than 1 level from the reference BITSS type for the corresponding photograph. The area of the circle represents the proportion of the classification represented. Cross sign indicates that there is no data here.

**Figure 3 nutrients-16-00855-f003:**
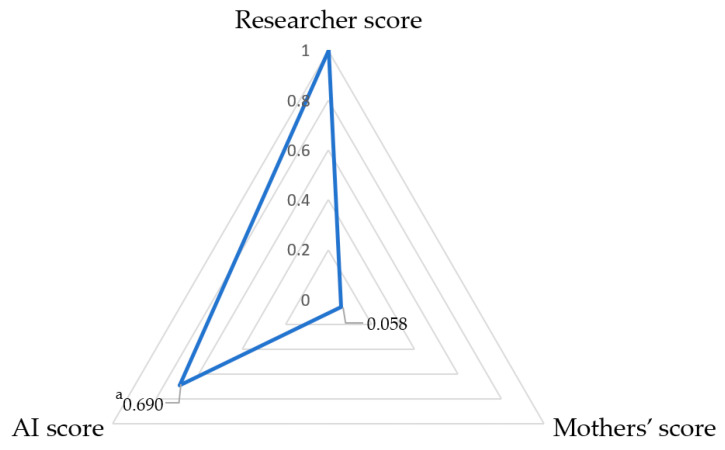
Radar chart of Kendall’s tau between AI, mothers’ score, and standard score (*N* = 905). ^a^: correlation coefficient, *p* < 0.001.

**Figure 4 nutrients-16-00855-f004:**
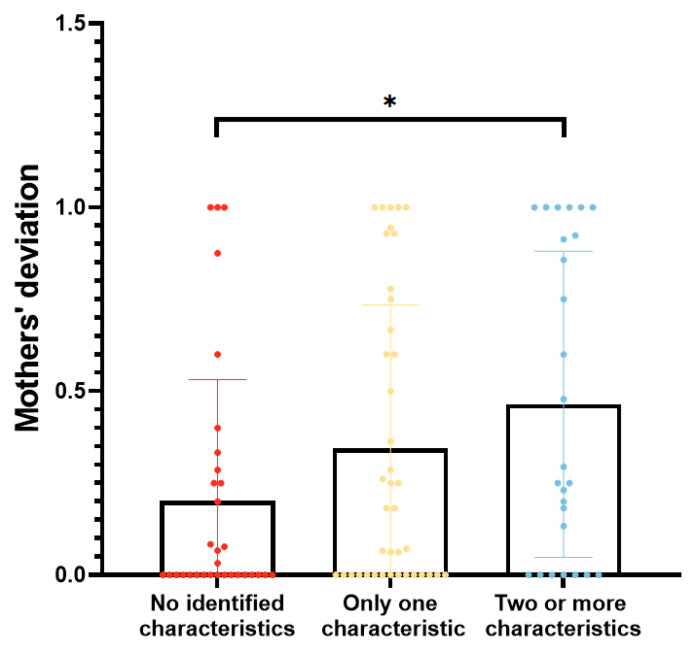
Comparison of parental deviation among different groups. *: comparison between groups with no identified features and with two or more features, *p* < 0.05.

**Table 1 nutrients-16-00855-t001:** Baseline characteristics of the study population.

Characteristics	Classification	*N* (%)
Age of mother (years)	<30	51 (41.2)
	30–35	59 (45.0)
	>35	18 (13.8)
Occupation	Employed	117 (89.3)
	Unemployed/Housewife	14 (10.7)
Education level	Junior college and below	43 (32.8)
	University and above	88 (67.2)
Per capita monthly income (yuan/month)	<6250	30 (22.9)
	6250–12,500	65 (49.6)
	>12,500	36 (27.5)
Parity	Multiparity	51 (38.9)
	Primiparity	80 (61.1)
Pregnancy complication history	Yes	78 (59.5)
	No	53 (40.5)
Mode of delivery	Vaginal	80 (61.1)
	Cesarean	51 (38.9)
Feeding patterns	Breastfeeding	72 (55.0)
	Mixed feeding	59 (45.0)
Postpartum depression	Yes	43 (32.8)
	No	88 (67.2)
EPDS score *		7 (4, 10)

* Median (P25 and P75). EPDS, Edinburgh Postnatal Depression Scale.

**Table 2 nutrients-16-00855-t002:** Agreement of stool classification between AI, mothers, and researchers.

Classification	Comparator Groups	ICC (95% CI)	Kendall’s W	Percentage Agreement (95% CI)
Seven types	AI vs. researchers	0.754 (0.719, 0.784)	0.836 *	95.5 (94.1, 96.9)
	Mothers vs. researchers	-	-	-
Four categories	AI vs. researchers	0.782 (0.752, 0.809)	0.840 *	95.8 (94.5, 97.1)
	Mothers vs. researchers	0.070 (−0.039, 0.169)	0.523	66.9 (63.8, 70.0)

ICC, intraclass correlation coefficients; Kendall’s W, Kendall’s coefficient of concordance W; CI, confidence interval; * *p* < 0.001.

**Table 3 nutrients-16-00855-t003:** Comparison of the degree of parental deviation with different characteristics.

Characteristic	Classification	Degree of Mothers’ Deviation, M (P25, P75)	Z/H	*p*
Age of mother (years)	<30 (*n* = 37)	0.00 (0.00, 0.44)	4.305	0.116
	30–35 (*n* = 44)	0.20 (0.00, 0.90)		
	>35 (*n* = 17)	0.29 (0.00, 0.83)		
Occupation	Employed (*n* = 86)	0.07 (0.00, 0.62)	−1.733	0.083
	Unemployed/Housewife (*n* = 12)	0.54 (0.19, 0.88)		
Education level	Junior college and below (*n* = 34)	0.25 (0.05, 0.92)	−2.047	0.041
	University and above (*n* = 64)	0.05 (0.00, 0.60)		
Per capita monthly income (yuan/month)	<6250 (*n* = 21)	0.07 (0.00, 0.80)	1.629	0.443
	6250–12,500 (*n* = 51)	0.23 (0.00, 0.86)		
	>12,500 (*n* = 26)	0.07 (0.00, 0.37)		
Parity	Multiparity (*n* = 40)	0.08 (0.00, 0.55)	−0.827	0.408
	Primiparity (*n* = 58)	0.23 (0.00, 0.76)		
Pregnancy complication history	Yes (*n* = 60)	0.24 (0.00, 0.73)	−0.324	0.746
	No (*n* = 38)	0.07 (0.00, 0.68)		
Mode of delivery	Vaginal (*n* = 65)	0.07 (0.00, 0.45)	−1.743	0.081
	Cesarean (*n* = 33)	0.26 (0.00, 0.94)		
Feeding patterns	Breast-feeding	0.22 (0.00, 0.77)	−1.249	0.212
	Mixed-feeding	0.03 (0.00, 0.53)		
Postpartum depression	Yes (*n* = 30)	0.33 (0.00, 0.94)	−1.719	0.086
	No (*n* = 68)	0.07 (0.00, 0.48)		
EPDS score *		7.00 (4.00, 12.00)	0.248	0.014

* presented as Spearman’s rank correlation coefficient (rho). EPDS, Edinburgh Postnatal Depression Scale.

**Table 4 nutrients-16-00855-t004:** Tobit regression analysis of factors influencing the degree of mothers’ deviation.

Characteristic	*β* (95% CI)	SE	*z*	*p*
Constant	−0.196 (−0.962, 0.570)	0.391	0.502	0.616
EPDS score	0.023 (0.001, 0.045)	0.011	2.077	0.038
Education level (ref. = Junior college and below)				
University and above	−0.149 (−0.403, 0.104)	0.129	−1.154	0.248
Mode of delivery (ref. = Vaginal)				
Cesarean	0.151 (−0.098, 0.399)	0.127	1.190	0.234
Occupation (ref. = Employed)				
Unemployed/Housewife	0.205 (−0.153, 0.562)	0.182	1.123	0.262

EPDS, Edinburgh Postnatal Depression Scale; CI, confidence interval.

**Table 5 nutrients-16-00855-t005:** Stratified Tobit regression analysis of the relationship between EPDS score and mothers’ deviation.

Stratified Condition	EPDS Score	Mothers’ Deviation	*β* (95% CI)	SE	*z*	*p*
Junior college and below ^1^	9.00 (4.75, 12.25)	0.25 (0.05, 0.92)	0.025 (−0.002, 0.051)	0.014	1.824	0.068
University and above ^1^	6.50 (4.00, 9.00)	0.05 (0.00, 0.60)	0.028 (−0.007, 0.063)	0.018	1.590	0.112
Cesarean ^2^	8.00 (4.00, 12.00)	0.26 (0.00, 0.94)	0.040 (0.025, 0.075)	0.018	2.262	0.024
Vaginal ^2^	7.00 (4.00, 10.00)	0.07 (0.00, 0.45)	0.014 (−0.015, 0.044)	0.015	0.955	0.339
Employed ^3^	7.00 (4.00, 10.50)	0.07 (0.00, 0.62)	0.024 (0.000, 0.048)	0.012	1.960	0.050
Unemployed/Housewife ^3^	8.50 (2.50, 12.00)	0.54 (0.19, 0.88)	0.036 (−0.011, 0.082)	0.024	1.508	0.132

EPDS, Edinburgh Postnatal Depression Scale; CI, confidence interval. ^1^ Adjusted the mode of delivery and occupation. ^2^ Adjusted the education level and occupation. ^3^ Adjusted the mode of delivery and education level.

## Data Availability

The data presented in this study are available on request from the corresponding author. The data are not publicly available due to privacy.

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
