# Peer review of "The Effectiveness of Artificial Intelligence in Assisting Mothers with Assessing Infant Stool Consistency in a Breastfeeding Cohort Study in China"

_nutrients, 2024, doi:10.3390/nu16060855_

Round 1

Reviewer 1 Report

Comments and Suggestions for Authors

An interesting manuscript evaluating stool analysis to understand pediatric gastrointestinal health. The authors have performed a study to evaluate stool of one-month-old infants’ feces on diapers based on photographs taken by a smartphone app. The photographs were independently categorized by a) Artificial Intelligence (AI), b) parents, and c) researchers/experts.

The manuscript is in general well designed, written and presented and, definitely, interesting for the journal audience.

A few comments that may help to improve the manuscript.

1.      In the abstract, please spell EPDS the first time that occurs. In general, please spell all abbreviations the first time that occur.

2.      The conclusion “AI-based stool evaluation service can effectively assist …” is rather too optimistic, since the κ= 0.636 for the AI shows a good agreement but not a perfect agreement. Propose to replace the phrase such as “AI-based stool evaluation service has the potential to assist …” or something similar…..

3.      Since there is a scale of stool rating, propose to replace the Cohen κ coefficient with a more appropriate concordance indicator, specifically the Kendall W coefficient, this coefficient takes to account the ordinal nature of data and is expected to show more accurately the agreement.

Author Response

Summary

Thank you very much for taking the time to review this manuscript. Please find the detailed responses below and the corresponding revisions in the re-submitted files.

Comments 1: In the abstract, please spell EPDS the first time that occurs. In general, please spell all abbreviations the first time that occur.

Response 1: Thank you for pointing this out. We have expounded any abbreviation when it appears for the first time to enhance reader comprehension. Specifically, in the abstract, the initial instance of "EPDS" has been revised to "Edinburgh Postnatal Depression Scale (EPDS)". (Refer to Page 1, Paragraph 1, and line 26).

Comments 2: The conclusion “AI-based stool evaluation service can effectively assist …” is rather too optimistic, since the κ= 0.636 for the AI shows a good agreement but not a perfect agreement. Propose to replace the phrase such as “AI-based stool evaluation service has the potential to assist …” or something similar...

Response 2: Your suggestion concerning the conclusion presentation is apt. To avoid exhibiting undue optimism, we have revised the phrase "AI-based stool evaluation service can effectively assist..." to "AI-based stool evaluation service has the potential to assist...". (Refer to Page 1, paragraph 1, and line 28).

Comments 3:  Since there is a scale of stool rating, propose to replace the Cohen κ coefficient with a more appropriate concordance indicator, specifically the Kendall W coefficient, this coefficient takes to account the ordinal nature of data and is expected to show more accurately the agreement.

Response 3: We agree with this comment. We have introduced the Kendall's W coefficient to replace the Cohen κ coefficient to measure the degree of agreements amongst AI, mothers, and standard scores and have re-analyzed the relevant data accordingly. The related modifications can be found in several locations throughout the revised manuscript, specifically: Page 4, Paragraph 6, and Line 171-172; Page 4, Paragraph 6, and Line 176-181; Page 6, Paragraph 4, and Line 239-240; Page 6, Paragraph 5, and Line 245; Page 6, Table 2, and Line 261-262; Page 10, Paragraph 2, and Line 344.

Furthermore, we also introduced the Kendall's tau to replace the spearman correlation coefficient, which is more suitable for ordinal data, the related modifications can be found in several locations throughout the revised manuscript, specifically: Page 5, Paragraph 1, and Line 185; Page 6, Paragraph 5, and line 251-252; Page 8, Figure 2, and Line 272; Page 10, Paragraph 2, and Line 345.

Reviewer 2 Report

Comments and Suggestions for Authors

The authors carried out a sensitive study where photographs of one-month-old infants’ feces on diapers were taken via a smartphone app and independently categorized by Artificial Intelligence (AI), parents, and re-searchers. The accuracy of the evaluation of the AI and the parents was assessed and compared.Whilst the study appears very interesting and captivating, the reviewer has a number of concerns, which need to be addressed:
a) The introduction needs more information that links breast milk, the baby and the stool even before stool consistency. So please, in the introduction, provide the digestive pathway of breast milk ingested to any given infant, and how it translates to fecal outcomes, prior to the consistency. Also, in the introduction, talk about the nutritional benefits of the breastmilk and how it reflects on the stool consistency. Provide a flow diagram to help support the various pathways.
The rationale/justification of this research is not so clear...please, where is the gap, afre they studies(or even one study) that has used photographs to evaluate fecal samples as means of evaluating infants? Please demonstrate the rationale
b) The materials and methods, please start this section with a new sub-section "Schematic overview of the cohort study", and should comprise 4 sentences, with a flow diagram showing the various stages. Please, try to connect this flow diagram's description with the objective of this work.
c) Results, please, try to remove the discussion in the result section and move them to the discussion. Results are supposed to relay the results, "what" and "where" only. The remaining "how", and "why"the results are behaving in this manner should be moved to discussion.
Please, try to make sure that in the discussion, use "(Refer to Table x)" or "(Refer to Figure x)" in all the places where data featured in results section are being discussed. That is to say, all the Tables and Figure in the results section must be featured in the discussion.

d) Please, your conclusions s too scanty. Provide more reflections of your study, and some recommendations/direction for future study. `

look forward to your revised manuscript

Author Response

Summary

Thank you very much for taking the time to review this manuscript. Please find the detailed responses below and the corresponding revisions in track changes in the resubmitted files.

Comments 1: The introduction needs more information that links breast milk, the baby and the stool even before stool consistency. So please, in the introduction, provide the digestive pathway of breast milk ingested to any given infant, and how it translates to fecal outcomes, prior to the consistency. Also, in the introduction, talk about the nutritional benefits of the breastmilk and how it reflects on the stool consistency. Provide a flow diagram to help support the various pathways.
The rationale/justification of this research is not so clear...please, where is the gap, afre they studies(or even one study) that has used photographs to evaluate fecal samples as means of evaluating infants? Please demonstrate the rationale.

Response 1: Thank you for pointing this out. We have expanded this section to elucidate the composition and digestive characteristics of the breast milk and its impact on fecal outcomes. We have added the phrase" Its composition, rich in lactose and unique fats, is more easily digestible and absorbable than formula milk. When lactose is not fully absorbed in the small intestine, it ferments in the colon, increasing stool water content and making it softer. Additionally, breast milk's oligosaccharides enhance beneficial gut bacteria, such as Bifidobacteria and Lactobacillus, further influencing stool consistency through fermentation and water content regulation ". (Refer to Page 1, Paragraph 2, and line 37-42). Corresponding references were cited, and serial number of subsequent references were adjusted accordingly. And the possible pathways by which breast milk affects infant fecal consistency also been presented in the graphical summary.

We appreciate your suggestion for the rationale of our study. While previous studies have employed photographic evaluation of fecal samples (referenced as 25 and 26 in our manuscript), However, they did not focus on breastfeeding infants to address the specific challenges in their stool consistency evaluation. Nowadays, AI technology offers the possibility of automated and quick fecal consistency assessment in large-scale breastfeeding cohort studies and in the research on time-varying relationships between breastfeeding and stool consistency. Therefore, we applied the AI-based stool evaluation service in an observational cohort study. Then in this study, we assessed the AI-graded scores and the mother-reported scores for early infant feces, and also identified those who could benefit most from this AI service. We have clarified and expanded on this rationale in the last paragraph of our introduction, highlighting the rationale and significance of our research. (Refer to Page 2, Paragraph 3, and line 73-75).

Comments 2: The materials and methods, please start this section with a new sub-section "Schematic overview of the cohort study", and should comprise 4 sentences, with a flow diagram showing the various stages. Please, try to connect this flow diagram's description with the objective of this work.

Response 2: Thank you for your valuable suggestion to introduce a new subsection titled "Schematic Overview of the Cohort Study" at the beginning of the Materials and Methods section. We have added the textual description and a flow diagram depicting the various stages of our study design and the connection between the whole cohort and the objective of this work within the newly created subsection. (Refer to Page 2, Paragraph 4, and line 80-87; Page 3, Figure 1, line 89). Corresponding, the relevant content covered in the old “study design” section also has been adjusted. (Refer to Page 3, Paragraph 1, and line 90-96; Page 3, Paragraph 2, and line 107-110).

Comments 3: Results, please, try to remove the discussion in the result section and move them to the discussion. Results are supposed to relay the results, "what" and "where" only. The remaining "how", and "why" the results are behaving in this manner should be moved to discussion.

Please, try to make sure that in the discussion, use "(Refer to Table x)" or "(Refer to Figure x)" in all the places where data featured in results section are being discussed. That is to say, all the Tables and Figure in the results section must be featured in the discussion.

Response 3: Thank you for your detailed read and deep analysis of our paper. We understand and agree with your suggestion that the results section should strictly relay the "what" and "where" of our findings, while the "how" and "why" should be moved to the discussion. Based on your suggestion, we 've rearranged the structure of the paper. The explaining and discussing of the results in results section have been moved to the discussion section. We 've also made sure to refer to all relevant data, tables, and figures in the discussion section, for seamless understanding and traceability of our results by readers. (Refer to Page 5, Paragraph 2, and line 190-193; Page 6, Paragraph 3, and line 235-236; Page 6, Paragraph 4, and line 240-243; Page 6, Paragraph 5, and line 247; Page 6, Paragraph 6, and line 254-257; Page 8, Paragraph 1, and line 275-276; Page 9, Paragraph 2, and line 297; Page 9, Paragraph 3, and line 306-309; Page 10, Paragraph 1, and line 327,331; Page 10, Paragraph 2, and line 337-338,345; Page 10, Paragraph 3, and line 352-353,355; Page 10, Paragraph 4, and line 360,362; Page 11, Paragraph 1, and line 363-367,387,389; Page 11, Paragraph 2, and line 395,400).

Comments 4: Please, your conclusions s too scanty. Provide more reflections of your study, and some recommendations/direction for future study. 

Response 4: We appreciate your insightful suggestion regarding the presentation of conclusions. The application of this AI service could be expanded to cover a broader population, further corroborating the accuracy of AI in stool consistency assessment. Therefore, we have expounded the conclusion to include: "Future work is needed to evaluate the applicability and effectiveness of this AI service in a broader population and in more complex feeding situations." And minor adjustments have also been made for some statements in the conclusion. (Refer to Page 12, Paragraph 1, and line 416-420).
